# Cellulose Nanocrystals Crosslinked with Sulfosuccinic Acid as Sustainable Proton Exchange Membranes for Electrochemical Energy Applications

**DOI:** 10.3390/membranes12070658

**Published:** 2022-06-26

**Authors:** Olena Selyanchyn, Thomas Bayer, Dino Klotz, Roman Selyanchyn, Kazunari Sasaki, Stephen Matthew Lyth

**Affiliations:** 1Department of Automotive Science, Graduate School of Integrated Frontier Sciences, Kyushu University, 744 Motooka, Nishi-ku, Fukuoka 819-0395, Japan; olenaselyanchyn@gmail.com; 2Lloyd’s Register Group Limited, Queens Tower A10F. 2-3-1, Minatomirai, Nishi-ku, Yokohama 220-0012, Japan; bayerthomas@gmx.net; 3International Institute for Carbon-Neutral Energy Research (WPI-I2CNER), Kyushu University, 744 Motooka, Nishi-ku, Fukuoka 819-0395, Japan; dino.klotz@i2cner.kyushu-u.ac.jp (D.K.); sasaki.kazunari.278@m.kyushu-u.ac.jp (K.S.); 4Kyushu University Platform for Inter/Transdisciplinary Energy Research (Q-PIT), 744 Motooka, Nishi-ku, Fukuoka 819-0395, Japan; 5Research Center for Negative-Emissions Technologies (K-NETs), Kyushu University, 744 Motooka, Nishi-ku, Fukuoka 819-0395, Japan; 6Next-Generation Fuel Cell Research Center (NEXT-FC), Kyushu University, 744 Motooka, Nishi-ku, Fukuoka 819-0395, Japan; 7International Research Center for Hydrogen Energy (HY30), Kyushu University, 744 Motooka, Nishi-ku, Fukuoka 819-0395, Japan; 8Department of Mechanical Engineering, University of Sheffield, Western Bank, Sheffield S1 3JD, UK

**Keywords:** nanocellulose, bionanomaterial, crosslinking, sulfonic acid groups, proton conductivity, polymer electrolyte membrane, fuel cell, low-cost PEM

## Abstract

Nanocellulose is a sustainable material which holds promise for many energy-related applications. Here, nanocrystalline cellulose is used to prepare proton exchange membranes (PEMs). Normally, this nanomaterial is highly dispersible in water, preventing its use as an ionomer in many electrochemical applications. To solve this, we utilized a sulfonic acid crosslinker to simultaneously improve the mechanical robustness, water-stability, and proton conductivity (by introducing -SO_3_^−^H^+^ functional groups). The optimization of the proportion of crosslinker used and the crosslinking reaction time resulted in enhanced proton conductivity up to 15 mS/cm (in the fully hydrated state, at 120 °C). Considering the many advantages, we believe that nanocellulose can act as a sustainable and low-cost alternative to conventional, ecologically problematic, perfluorosulfonic acid ionomers for applications in, e. fuel cells and electrolyzers.

## 1. Introduction

Proton exchange membranes (PEMs) are an essential component in proton exchange membrane fuel cells (PEMFCs) and proton exchange membrane water electrolyzers (PEMWEs), both of which are key technologies contributing to the establishment of a hydrogen society [1]. Fuel cells are electrochemical devices which can transform chemical energy into electrical energy with high efficiency, via a controlled reaction between hydrogen and oxygen [2]. The function of the electrolyzer is the same, but in reverse—producing hydrogen and oxygen from water, using electrical energy. In PEMFCs, the PEM facilitates the diffusion of hydrogen ions (i.e., protons) from anode to cathode, provides a mechanical support for the electrodes, and acts as a selective barrier to prevent fuel, oxidants, or electrons from passing through. In PEMWEs, the PEM performs essentially the same tasks, as well as keeping the generated gases separate [3].

The commercialization of PEMFCs and PEMWEs is progressing, but the pace should be further accelerated to achieve decarbonization by 2050. One factor slowing the penetration of electrochemical systems into a crowded marketplace is the high cost of the materials, a large proportion of which is due to the PEM [3,4]. Currently, Nafion^®^ is the benchmark polymer used in PEMs. This is a sulfonated fluoropolymer with high proton conductivity (~100 mS/cm in the hydrated state), which has dominated the market since its discovery in the late 1960s [5]. The high proton conductivity in Nafion^®^ is reportedly due to the phase separation between the hydrophobic and hydrophilic domains, generating percolating ion conducting channels that deliver efficient proton transport at a relatively low ion exchange capacity [6]. However, Nafion^®^ is extremely expensive due to the complex manufacturing processes, as well as being difficult to recycle. An additional drawback, relevant to direct methanol fuel cells (DMFC) and flow batteries, is the high permeation rates of methanol [7] or other active species [8], resulting in crossover issues and fuel loss. As such, more sustainable, lower cost, and low permeability alternatives to Nafion^®^ could be highly beneficial. Numerous researchers have investigated various proton-conducting polymers that could compete with Nafion^®^. Regarding sustainability and cost, a promising class of materials are sulfonated hydrocarbon PEMs, such as sulfonated poly(ether ether ketone) (SPEEK), sulfonated poly(aryl ether sulfone) (SPAES), sulfonated polyimide (SPI), sulfonated poly(phenylene), and branched sulfonated poly(ether ketone) (SPEK) [9]. However, most of the hydrocarbon PEMs are derived from petrochemicals and, therefore, still rely on the extraction of fossil fuels, and expensive fabrication processes.

Sustainability and the circular economy are central to the next phase of industrial practice, driven by factors including sustainable growth, energy security, the carbon footprint, and efficient resource management [10]. As an alternative to petrochemicals, biomaterials and biopolymers proved to be a more sustainable alternative. Cellulose is the main constituent of plants and trees, representing the most abundant material resource in the world [11], and is arguably the most sustainable resource available to society today. As a sub-class of cellulose, nanostructured forms of cellulose (collectively referred to as nanocellulose) can be produced by the processing of conventional cellulose, and are now being widely considered for energy applications [12,13]. There are three major classes of nanocellulose, namely: cellulose nanocrystals (CNC, sometimes also referred to as nanocrystalline cellulose); cellulose nanofibers (CNF, sometimes also referred to as nano fibrillated cellulose); and bacterial cellulose (BC). These all differ significantly in their microstructure, the method of production, and in their materials’ properties [13].

Nanocellulose is attractive as an alternative material for PEMs due to its intrinsic properties, such as high mechanical strength [14], exceptional gas barrier properties [15], acidic functional groups leading to proton conductivity [16], and a polymer structure suitable for various chemical modifications. As a biodegradable polymer, it has an additional advantage over the fluorinated PEMs and hydrocarbon PEMs, such as Nafion^®^, in terms of sustainability and the circular economy. Nanocellulose-based PEMs could have specific advantages in terms of short-term applications, such as single-use, or disposable devices.

Several research groups have employed different types of nanocellulose materials in PEMs. The vast majority of the existing literature describes the utilization of BC, CNC, or CNF primarily as a reinforcing agent mixed with established ionomers, predominantly Nafion^®^ [12,17]. However, this approach does not solve the issues associated with using perfluorinated polymers, only slightly reducing the overall amount used. Alternatively, the exploration of nanocellulose as the primary material in a PEM may provide a means of completely avoiding the use of perfluorinated polymers.

In previous work, we demonstrated that the PEMs fabricated from pure CNF and CNC possess intrinsic proton conductivity (0.05 and 4.6 mS/cm, respectively), but the values measured were much lower than the benchmark ionomers, such as Nafion^®^ [16]. Another important finding was that the nanocellulose PEMs displayed remarkably low hydrogen permeability compared to Nafion^®^, in principle enabling the fabrication of much thinner membranes in which the lower specific resistance can compensate for the lower proton conductivity [16,18].

Several research groups have succeeded in improving the macro-mechanical properties of cellulosic materials, using chemical crosslinking. The simplest method involves a one-step reaction between the hydroxyl groups of cellulose and the carboxylic acid groups of an organic acid with two or more acid moieties, resulting in the formation of ester bonds. Succinic acid (two carboxylic acid groups), citric acid (three acid groups), and butanetetracarboxylic acid (four acid groups) were reported as crosslinkers in conventional cellulosic materials, mainly for improving the mechanical properties [19,20,21].

Recently, citric acid has been demonstrated as a cross-linking agent for Cladophora algae-derived nanocellulose membranes, resulting in improved water-stability and increased pore size, for the nanofiltration of gold nanoparticles [22]. In another study, succinic acid was reported to reinforce BC agar, resulting in an improvement of the mechanical properties and lowered water sorption [23]. However, applying the above crosslinkers to PEMs is expected to decrease the proton conductivity, because the weakly acidic hydroxyl groups will be sacrificed as sites for esterification.

Addressing this, the combination of crosslinking and sulfonation has been previously applied to simultaneously improve proton conductivity in polymers, as well as improving membrane integrity or aqueous stability. For example, polyvinyl alcohol (PVA) has been crosslinked with sulfosuccinic acid (SSA), an organic acid with two pendant carboxylic acid moieties and one central sulfonic acid moiety (-SO_3_^−^H^+^) (Figure 1). In one study, PVA crosslinked with SSA achieved an in-plane proton conductivity of up to 10 mS/cm at room temperature [24]—approximately a 10-fold increase compared to pure PVA [25]. Sulfosuccinic acid has also been applied as a crosslinker in microcrystalline cellulose fibers, resulting in a maximum in-plane proton conductivity of 40 mS/cm at 80 °C [26], or, with CNF as the starting material, achieving through-plane proton conductivity of 3.17 mS/cm [27]. The latter work reported a worsening of the mechanical properties and crystallinity of the cellulose membranes, due to damage of the amorphous regions by the acid.

In this work, we explore the use of SSA as a crosslinking agent to improve the proton conductivity and aqueous stability of nanocellulose-based PEMs. However, we take the above works a step further. For the first time, we demonstrate the crosslinking of CNC for PEM applications, a cellulose variant which is often overlooked as a PEM, due to its very high dispersibility in water. The primary advantage of using CNC is that, as a higher purity material, it can support a higher proportion of the sulfonic acid groups within its chemical structure compared to CNF and BC, potentially resulting in higher proton conductivity. In addition, CNC is a crystalline material formed by the removal of amorphous portions of microcrystalline cellulose via strong acid hydrolysis. As such, CNC is expected to be much more resistant to the degradation of the polymer backbone during crosslinking with SSA. Once the problem of the aqueous stability is solved, the crosslinked CNC membranes could represent low-cost and sustainable PEMs with reasonable proton conductivity and mechanical strength, fulfilling the critical requirements for use in both PEMFC and PEMWE systems.

## 2. Materials and Methods

### 2.1. Nanocellulose Membrane Preparation

The commercially available CNC powder used in this study (CelluForce NCC^®^, Montreal, QC, Canada) comprises needle-shaped crystals with nominal particle sizes of 7.5 × 150 nm, a specific surface area of 550 m^2^/g, a degree of crystallinity of 89.9%, and a bulk density of 0.7 g/cm^3^ [28]. Since this CNC powder was produced using 64 wt.% sulfuric acid hydrolysis, the chemical structure contains residual -SO^3^_+_ functionalities, usually in neutral (salt) form [28,29]. The as-received CNC powder was first dispersed in deionized water to obtain a 1 wt.% solution, with the aid of a high-shear mixer (1000 rpm, AS-ONE, Osaka, Japan). Sulfosuccinic acid (SSA, 70 wt.%, Sigma Aldrich, Tokyo, Japan) was then diluted to 5 wt.% in deionized water, and combined with the CNC dispersion in different ratios, using ultrasonic homogenization (5 min, Branson SFX 250). The resulting mixtures were denoted as CNC–x%–SSA, where x is the weight percentage of SSA relative to CNC (x = 3 to 50 wt.%). After homogenization, the resulting dispersions were cast into polytetrafluoroethylene (PTFE) dishes and heated in a convection chamber at 35 °C for 12 h until completely dry. Crosslinking was performed by increasing the temperature in the same convection chamber to 120 °C for 10 min. The PTFE dishes were then removed from the chamber and cooled to room temperature. After this, the fabricated membranes were thoroughly washed with deionized water to remove physically adsorbed and unreacted SSA from the membranes. The crosslinking reaction between the CNC and SSA is depicted in Figure 1. According to earlier reports, the main reaction pathway involves the formation of ester bonds between the carboxylic groups of SSA and the hydroxyl groups of nanocellulose [28].

### 2.2. Mechanical Properties 

The tensile strength, elastic modulus, and elongation at the break of the pristine cellulose and acid-crosslinked membranes were measured using a texture analyzer (EZ-SX, Shimadzu, Japan), under ambient conditions (~25 °C, ~70% relative humidity). Dumbbell shape samples (JIS K6251-7, ISO 37-4 standard) were cut from the larger membranes using a punch (Dumbbell Co., Ltd., Kawagoe, Saitama, Japan), and several of these were tested for each membrane to ensure reproducibility. The samples were clamped in the analyzer, and the load (N) and elongation (mm) were recorded with an extension velocity of 1 mm per minute. The data were recorded using material testing software (TRAPEZIUM X, V 1.4.0, Shimadzu Ltd., Kyoto, Japan). The sample thickness was measured prior to the test, using a digital micrometer (Mitutoyo, Kawasaki, Kanagawa, Japan).

### 2.3. Proton Conductivity

The ionic conductivity of the fabricated membranes was studied over a range of temperatures and relative humidity, using a membrane testing device (MTS-740, Scribner Associates, Southern Pines, NC, USA) coupled with an impedance spectroscopy analyzer (ModuLab XM MTS, Ametek, Tokyo, Japan). The impedance spectra were acquired with an AC amplitude of 10 mV in a frequency range from 1 MHz to 10 Hz. In a standard test, the impedance was measured at a stable temperature and isothermal changes of relative humidity (RH) from 20% RH to 95% RH. Before the acquisition, the membrane was pretreated at 70% RH for 1 h, followed by 15 min pretreatment at each successive humidity (the conditions recommended by the MTS740 manufacturer) [30]. For selected membranes, the impedance was measured from 40 to 120 °C to study the influence of temperature on the proton conductivity. At temperatures above 100 °C, the measurement chamber was back-pressurized at 230 kPa. To extract the membrane resistances R (Ω), the spectra in the high frequency region were fitted to an equivalent circuit, using ZView^®^ software (Scribner Associates). The membrane resistance was determined from the high frequency intercept or equivalent circuit fitting, and the through-plane conductivity σ (mS/cm) was calculated using the standard procedure [30], using the following equation:(1)σ=L/(R·A)
where *L* is the membrane thickness (*L*, cm); and *A* is the cross-sectional area (0.5 cm^2^).

### 2.4. Ion Exchange Capacity 

To measure the ion exchange capacity (IEC), the membranes’ samples were first dried in a vacuum chamber and then immediately weighed and their dry weight recorded. Then, the membranes were placed in 50 mL of 1 mol/L sodium chloride solution for 24 h, to allow the protons to be completely exchanged with the sodium ions. The membranes were then removed from the solution, and the remaining sodium chloride solutions titrated with a 0.01 mol/L solution of sodium hydroxide. The endpoint volume of the titration was estimated from the titration curves by measuring the pH of the solution, using a high precision digital pH meter (F-71 with electrode 9618S-10D, HORIBA, Kyoto, Japan). The number of protons extracted from the membrane was estimated and used to calculate the IEC by accounting for the dry mass of the membrane.

### 2.5. Membrane Characterization 

The insights into the chemical structure of the membranes were obtained via scanning Fourier transform infrared (FTIR) spectroscopy (Nicolet iN10 MX), in attenuated total reflectance (ATR) mode, with a scan range of 4000 to 650 cm^−1^. The chemical composition of the composite membranes was also determined, using X-ray photoelectron spectroscopy (XPS, PHI 5000 Versa Probe/Scanning ESCA microprobe with Mg Kα radiation, 12 kV, 10 mA). The surface and cross-sectional morphology of the membranes were observed using a field emission scanning electron microscope (FE-SEM, JSM-7900F, JEOL, Tokyo, Japan). The energy dispersive X-ray spectroscopy (EDS) analysis with elemental mapping was performed, using two detectors (Oxford Instruments, Abingdon, UK) coupled to the FE-SEM. To prevent sample charging under electron irradiation in the SEM chamber, the samples were coated with a thin layer of platinum, using an ion sputterer (Hitachi E-1030) before observation. The wide angle X-ray diffraction (XRD) measurements were performed using Cu Kα radiation (Rigaku Smartlab, Tokyo, Japan), and the data were collected from 5° to 50° at a scanning rate of 0.2°/min.

## 3. Results

### 3.1. Appearance and Morphology

The pristine nanocellulose membranes prepared by solution casting are generally transparent, due to the high crystallinity and small size of the crystallites, resulting in a reduced scattering of light, as well as iridescence when tilted under a light source [31]. Figure 2a shows photographs of the various CNC membranes fabricated by casting and then cut into 3 × 3 cm sized samples (more samples and tilted membranes’ photos are given in Appendix A). The pure CNC membrane is colorless and highly transparent, as expected. However, the crosslinked membranes display a systematic change in color to dark brown and then black, and a corresponding decrease in transparency as the proportion of SSA increases. This provides a visual confirmation that the crosslinking reaction occurred successfully, even at a low SSA content (3–9%). The fabrication of the CNC membranes with SSA content up to 50 wt.% was possible, but above 40 wt.%, the formed membranes developed defects and cracks during crosslinking at 120 °C. As such, the maximum SSA content used herein is 35 wt.%. It is worth noting that, due to the nanostructure of the CNC, a much shorter time was sufficient for crosslinking (10 min) in contrast to the microcrystalline cellulose (2 h) as previously reported [26]. The color change is reportedly directly related to the crosslinking density (i.e., the amount of acid in the membrane-forming solution), and a similar color change was previously reported for CNF crosslinked with SSA [26].

The SEM observation of the surfaces of the membranes clearly reveals the nanocrystalline nature of the CNCs (Figure 2b). However, a clear difference is observed in the alignment of the crystallites. In the CNC membrane, there is a high degree of alignment, as commonly observed in cellulosic materials, due to the mutual attraction of individual crystallites via the plurality of hydrogen bonds within their structure. In contrast, almost no alignment is observed in the crosslinked membranes, with a random distribution of nanocrystals. To demonstrate the influence of SSA on this process, SEM images of the samples with a lower SSA content were obtained (Appendix A), showing the alignment of the CNC crystallites below around 3 wt.% SSA, and random orientation above around 5 wt.%. This is evidence that the presence of SSA disrupts the hydrogen bonding network of cellulose, preventing alignment. An important observation is that the crystallites retain their approximate physical dimensions, even after crosslinking. The preservation of the crystalline structure is expected to contribute to the retention of the mechanical properties of the membrane. The cross-sectional SEM observation (Figure 2c) reveals similar trends in the structure, with some alignment in the CNC membrane, and a more homogenous structure in the cross-linked membrane (CNC-25%-SSA).

### 3.2. Elemental Composition and Chemical Structure

#### 3.2.1. Elemental Mapping

To confirm the elemental composition of the crosslinked membranes, SEM-EDS was conducted (Figure 3). Despite the relatively low atomic content of sulfur in the membrane (theoretically ~2 wt.% in the CNC-25%-SSA membrane shown here), the S Kα1 signal was clearly detected, along with the other main chemical elements expected in the membrane (i.e., carbon and oxygen). Moreover, the signal distribution was homogenous, suggesting that the SSA was evenly distributed throughout the membrane. The isotropic structure and composition of the crosslinked membranes observed both by the SEM and EDS is considered to be beneficial for properties such as the proton conductivity, since PEMs with an anisotropic structure (e.g., graphene oxide) have been shown to result in different values of conductivity depending on the orientation of the measurement (i.e., in-plane vs. through-plane) [32].

#### 3.2.2. Infrared Spectroscopy

Several characterization techniques were conducted to verify that the structure of the crosslinked membranes was as envisioned. Figure 4 shows the ATR-FTIR spectra obtained for the pure CNC and the crosslinked CNC-SSA membranes with up to 40 wt.% SSA content. Multiple peaks are observed in the spectra, most of which were identified by comparison with the literature on cellulosic materials [33]. Several specific changes suggesting successful crosslinking are observed. Specifically, the OH stretching vibration peak at ~3300 cm^−1^ decreases in intensity, which is attributed to the formation of ester bonds between the -OH groups in the cellulose and carboxylic groups in SSA. Other major changes are observed at ~1716, ~1230, and ~1203 cm^−1^ associated with the formation of the ester bond, and the presence of the sulfonic moiety [34], respectively (highlighted in green and blue in the chemical structure, inset). Another minor but informative change occurs in the two peaks at 1057 and 1034 cm^−1^, corresponding to a decrease in the proportion of the C-O-H bonds, and an increase in the proportion of the C-O-C bonds, respectively, as expected upon the formation of ester bonds in the crosslinked structure.

#### 3.2.3. X-ray Diffraction

X-ray diffraction was performed to verify the effect of crosslinking upon the crystallinity, as depicted in Figure 5. All of the samples exhibited characteristic sharp peaks, ~15.4, 22.7°, and 34.5°, assigned to the (101), (200), and (040) lattice planes of the typical for cellulose I allomorph, respectively [28,35]. A small decrease in the (101) peak area is observed in the crosslinked samples compared to CNC (Appendix A), which could be attributed to the disorder observed in the microscopic morphology observed by SEM. Overall, the XRD results are remarkably similar for all of the samples, indicating that the crosslinking reaction does not affect the crystalline structure of CNC, despite taking place at a high temperature, under acidic conditions. This finding differs significantly from the results of the other studies using conventional cellulose [26], or CNFs crosslinked with SSA [27]. In both studies, the crystallinity decreased sharply as the amount of acid in the crosslinked samples was increased. In contrast, here, the utilization of the highly crystalline CNCs rather than the more amorphous CNFs enabled the preservation of the crystal structure to a much greater extent. This retention of the crystal structure is likely to be important from the perspective of the mechanical integrity of the membranes.

#### 3.2.4. X-ray Photoelectron Spectroscopy

X-ray photoelectron spectroscopy (XPS) survey spectra of the eight different membranes (Figure 6a) does not reveal any major changes in the elemental composition after crosslinking, and the C/O ratio does not significantly change (Appendix A). A magnified view of the S 2p region is shown in Figure 6b. The main peak at ~168.5 eV is typically assigned to the sulfone moiety [36], as expected after crosslinking with SSA. Interestingly, sulfur (0.6 wt.%) is detected in the pristine CNC sample, attributed to sulfonation via acid hydrolysis during the production process [16]. The intensity of this sulfur peak increases slightly after crosslinking, with values reaching 1.16 wt.% (although these values are close to the detection limit of the technique, resulting in a high degree of error). The narrow XPS scans of the same region, shown in Appendix A, reveal a similar result, namely that the binding energy characteristic of the sulfonic moiety increases in intensity after crosslinking.

The narrow scans were performed in the C1s region to assess the changes in the structure of cellulose upon crosslinking. Figure 6c shows three representative samples: pristine CNC; CNC-10%-SSA; and CNC-25%-SSA, deconvoluted into four different chemical states: C-C (~284.5 eV); C-O (~286.1 eV); O-C-O (~287.5 eV); and C=O (~288.7 eV) [16,36]. Again, the samples have similar profiles, confirming that there is no dramatic change in the chemical structure of the cellulose backbone upon crosslinking, despite the major changes in the color and transmittance. The smallest peak located around 289 eV likely corresponds to carbon atoms bound to oxygen, both on the CNC surface and in the structure of the added crosslinker. This peak increases from a contribution of 1.5% in pristine CNC to 6–7% in the crosslinked samples. The C 1s profile is quite different compared with an earlier study on a different cellulosic source material [16], highlighting that the chemical structure of nanocellulose can vary significantly depending on the raw material, the provider, and the fabrication process. These differences are likely to have some influence on the material properties and should be carefully considered for practical applications.

### 3.3. Mechanical Properties

The extraordinary mechanical properties are one of the most attractive features of nanocellulose [12]. It has been reported that the elastic modulus of a single nanocellulose fibril can be as high as 50 GPa in the transverse direction, and up to 18.4 GPa in the longitudinal direction [14]. The stress–strain curves, Young’s modulus (MPa), the ultimate tensile strength (MPa), and elongation at break of pristine CNC and the crosslinked CNC-x%-SSA membranes are shown in Figure 7.

The stress–strain curves of pristine CNC and CNC-35%-SSA membranes are shown in Figure 7a,b. The CNC membrane is relatively inelastic, reaching only ~1% elongation at break. On the other hand, the crosslinked sample with 35 wt.% SSA is much more elastic, achieving over 6% elongation at break. This significant improvement in the elastic behavior is proportional to the amount of crosslinker used, as summarized in Figure 7c. Conversely, the Young’s modulus decreases from 5.8 GPa for pristine CNC to ~1.7 GPa for CNC-35%-SSA, also proportional with the amount of crosslinker (Figure 7d). In general, the crosslinking leads to an increased stiffness in polymeric materials, for example, in the case of starch crosslinked with citric acid [37]. This effect was not observed in previous studies on SSA crosslinked cellulose, attributed to chemical damage of the cellulose backbone. Here, significant damage to the cellulose is not observed, due to the high crystallinity of CNC (as confirmed by XPS and FTIR). However, there is a significant change in the alignment of the crystallites after crosslinking (Figure 2), attributed to the reduced hydrogen bonding between crystallites. This competition between the bonds likely leads to the observed changes, both in the elastic modulus and elongation. Meanwhile, the ultimate tensile strengths of the membranes vary between 25.9 MPa and 48.5 MPa, although no clear trend emerges with the amount of crosslinker. These results indicate that the crosslinking of crystalline nanocellulose can be used to tune the mechanical properties to reduce brittleness and increase the elasticity, which is an important factor considering its applications as a PEM.

Despite the decreases in Young’s modulus and the tensile strength after crosslinking, the values obtained for SSA-crosslinked CNCs are much higher compared to the benchmark PEMs, such as Nafion^®^, measured using the same equipment (Young’s modulus: 0.25 MPa, tensile strength: ~14 MPa; Appendix A). Moreover, a comparison with the related literature on crosslinked cellulosic materials (Table 1) shows that the type of cellulose used has a major impact on the mechanical properties. Using CNC as a starting material results in stronger membranes by several orders of magnitude compared with CNF [27], or conventional cellulose [26]. This is attributed to the observed decrease in crystallinity after crosslinking in these studies.

### 3.4. Water Stability and Ion Exchange Capacity

An essential feature of PEMs is that they should be stable in water, since they generally operate in humid conditions, and water is generated at the cathode. The pristine CNC membranes disintegrate and redisperse readily upon immersion in water (Appendix A). Meanwhile, the crosslinked membranes are indefinitely stable in water (Appendix A) under ambient conditions, providing further evidence for the success of the crosslinking reaction. The distinctive feature of the crosslinked membranes is that, despite a considerable water uptake (Appendix A), they show almost no lateral swelling in water (Appendix A), with most of the swelling (by 10–20%) taking place in the through-plane direction (Appendix A). 

The stability in water allowed for the ion exchange capacity (IEC) to be reliably measured using titration (titration curves for all of the samples are given in Appendix A). Figure 8 plots the measured IEC of nine different CNC-x%-SSA membrane formulations (where x = 10 to 50%). The dashed line shows the theoretical change in IEC depending on the proportion of SSA, assuming that the IEC of the pristine CNC is negligible, and that all of the SSA is chemically bonded to CNC. The experimental results correspond extremely closely with the calculated IEC, up to 30% SSA content (1.423 mmol/g). However, at higher concentrations, the IEC saturates and is lower than the predicted value. This strongly suggests that ~30% SSA is the maximum amount that can be fully accommodated in the membrane as a crosslinker. At higher concentrations, the residual acid is not likely to form ester bonds, remaining in the free state. For the membranes fabricated with higher SSA content (especially with 45 to 50% SSA content), the membrane integrity was noticeably poor after the crosslinking reaction at a high temperature. This is directly attributed to the presence of this excess SSA, resulting in damage to the membrane. As such, the samples with a high SSA content are not suitable for PEM application, and are herein discounted from further measurements.

### 3.5. Proton Conductivity

The values of the proton conductivity of the fabricated membranes were calculated using Equation (1), considering the individual membrane thicknesses and the resistances obtained from the impedance spectroscopy measurements. The representative through-plane impedance spectra of the pristine CNC and the crosslinked CNC-25%-SSA membranes are shown in Appendix A. The dependence of the proton conductivity on relative humidity and temperature for the pristine CNC and CNC-25%-SSA membranes is shown in Figure 9a,b. Both of the samples undergo a dramatic change in conductivity with relative humidity, with a variation of almost five orders of magnitude between the dry and fully hydrated states. This is typical behavior for PEMs, such as Nafion^®^, in which water acts as the primary charge transport medium [16]. In the case of CNC, the conductivity increases steadily from ~10^−5^ mS/cm at low RH to a maximum of ~0.4 mS/cm in the fully hydrated state. Increasing the temperature from 40 °C to 120 °C results in an increase in conductivity of around an order of magnitude, regardless of the relative humidity.

In the case of CNC-25%-SSA, the proton conductivity is consistently higher than that of the pristine CNC membrane. In this case, the increase in conductivity with relative humidity increases relatively rapidly at a lower RH, and slowly at a higher RH, up to ~12 mS/cm at 97% RH. Since the membranes containing SSA as crosslinker adsorb water from the environment faster than pure CNC (as shown in Appendix A) the humidification above 60% RH has less impact on the proton conductivity. In terms of temperature dependance for the crosslinked membrane, at a low RH there is a regular increase in conductivity from 40 °C to 120 °C, by around an order of magnitude. In contrast, at a high RH there is only a minor increase in conductivity between 60 °C and 120 °C. This is similar to the behavior of ionomers, such as Nafion^®^, in which the water acts as a charge transport medium and full hydration is more relevant to conductivity than temperature.

As such, there is a profound increase in the proton conductivity between the two membranes. This confirms that the inclusion of sulfonic acid groups in the crosslinker can successfully improve the conductivity, by around two orders of magnitude. In addition to the improved aqueous stability and the enhanced mechanical properties, this suggests that the crosslinked CNC membranes may be highly suited as PEMs in practical electrochemical applications. The fabricated membranes achieved better conductivity compared to the other purely nanocellulose membranes, for instance, carboxylated nanocellulose [38] σ_⊥_ = 1.5 mS/cm, a composite of bacterial cellulose and fucoidan [39] σ_⊥_ = 1.6 mS/cm, and also CNF crosslinked with SSA [27] σ_⊥_ = 3.17 mS/cm. In addition, we have demonstrated the stability of the membranes at temperatures as high as 120 °C, while mentioned studies reported the measurements below 100 °C.

### 3.6. Activation Energies

The activation energies were determined from the slopes of Arrhenius plots to gain an insight into the proton conduction mechanisms (Figure 9c,d). The activation energies for the CNC membrane were found to be largely independent of humidity, similar to CNC-25%-SSA at 30% and 70% RH, and also similar to the reported values for cellulose-based PEMs [39]. In terms of the proton conduction mechanism, it has been reported that the vehicle mechanism dominates at lower relative humidity, in which the protons diffuse as a relatively large water-solvated species (i.e., H_3_O^+^, H_5_O_2_^+^, and H_9_O_4_^+^) in tandem with a counter diffusion of unprotonated water molecules [9,16]. The vehicle mechanism is usually associated with higher activation energies, as actually observed in Figure 9c,d, with both CNC and CNC-25%-SSA having similar activation energy of around 30 kJ/mol at 30 and 70% RH.

Meanwhile, at a high relative humidity, the crosslinked samples demonstrate much lower activation energy (13.2 kJ/mol for CNC-25%-SSA at 97% RH), with values comparable to the benchmarks, such as Nafion^®^. Lower activation energy is associated with Grotthuss-like proton transport, which in this case may be related to the presence of local hydrated centers around the sulfonic groups, leading to a more complete hydrogen-bonding network, i.e., faster and more numerous conduction pathways [16].

Figure 9e,f shows the dependence of proton conductivity on the proportion of SSA crosslinker in the membranes, at 80 and 120 °C. A clear trend emerges, with the proton conductivity dramatically increasing in proportion with the SSA content up to 30 wt.%, slightly decreasing at 35 wt.% (at 80 °C and 95% RH). The same general trend is observed at 120 °C. This provides further evidence that SSA is successfully incorporated as a crosslinker, and that the provided -SO_3_H moieties participate in proton transport. The increase is also largely proportional to the measured ion-exchange capacity of the membranes. The highest conductivities of 10.4 and 14.0 mS/cm were achieved at 80 and 120 °C, respectively, for the CNC-30%-SSA membrane. The achieved values of conductivity are still more than one order of magnitude lower compared to the benchmark membranes, such as Nafion^®^ (>100 mS/cm in similar conditions). In comparison with the nanocellulose-based membranes, the conductivities are almost one order of magnitude higher (see Table 2). Specifically, the conductivity of the CNC-x%-SSA membrane exceeds that of the cellulose nanofiber membranes crosslinked with SSA [27], the carboxylated CNF membranes [38], as well as the composite bacterial cellulose-fucoidan-tannic acid membranes [39]. Further increasing the SSA content resulted in a decrease in conductivity, correlating with the decrease in IEC and the deterioration in the mechanical properties. The decrease may be explained similarly to the IEC decrease, namely inadequate acid accommodation as crosslinker, and damage of the CNC by free acid at high temperatures. As such, it is concluded that the CNC membranes with 25 to 35 wt.% SSA will be optimal for PEM applications, due to the combination of high conductivity, aqueous stability, and good mechanical properties.

### 3.7. Retention of Conductivity after Boiling

A key requirement of PEMs in industry is durability. For instance, the Fuel Cell Commercialization Conference of Japan requires that commercial PEMs should retain a proton conductivity of 95% of the initial value after boiling in water for 1000 h, as a test of the membranes’ stability [40]. As such, to estimate the durability of crosslinked cellulose, a CNC-35%-SSA membrane was placed into boiling water for 3 h. After boiling, the structural integrity of the membrane was preserved, and no color change was detected (unlike pristine CNC, which readily dissolves upon immersion in water, even at an ambient temperature, see Appendix A). This confirms the success of crosslinking in terms of improving the mechanical properties, even in the harsh environments that are characteristic of e.g., fuel cell operation. However, the pH of the water after the test was slightly acidic (pH ~ 4), suggesting that some of the SSA leaches from the membrane during boiling.

To confirm this, the conductivity of the CNC-35%-SSA membrane after boiling was also measured and compared with the initial conductivity (as well as the conductivity of a pristine CNC membrane) (Figure 10). After boiling, the conductivity decreased by a factor of around six. This is attributed to the SSA leaching from the membrane, possibly because of a de-esterification reaction. This result places some limitation on the application of the current membranes at high temperatures and/or humid conditions. In future, it may be beneficial to explore different crosslinking reactions avoiding ester bonds, to further enhance the durability. However, the conductivity is still significantly higher after boiling compared to the pristine CNC membrane. In addition, the stability of the membranes was checked in Fenton reagent (aq. solution of 3% H_2_O_2_ and 2 ppm Fe^2+^), and the membranes did not survive the treatment for 1 h at 80 °C. It is indeed a drawback for this material in its current form, and further improvements are required for future studies. 

## 4. Discussion

The structure of cellulose containing a high proportion of hydroxyl functional groups on the surface provides a convenient starting point for chemical modification. Recently, we combined chemical sulfonation of CNFs to increase the conductivity (from 0.05 to 2 mS/cm) with the fabrication of thin membranes (8 µm) to minimize the membrane resistance, achieving an industrially relevant power density of 156 mW/cm^2^ in a fuel cell setting [18]. However, the durability and mechanical stability of such membranes in a humid environment, and the much lower conductivity compared to Nafion^®^, remain as significant issues moving forward. In another approach, Guccini et al. reported PEMs (thicknesses of 14 and 24 µm) based on carboxylated CNF resulting in through-plane proton conductivity of up to ∼1.5 mS/cm at 30 °C and 95% RH (estimated from in situ fuel cell tests) [38].

Sulfosuccinic acid has also been applied as a crosslink in microcrystalline cellulose fibers, resulting in a maximum in-plane proton conductivity of 40 mS/cm at 80 °C [26]. Unfortunately, the above works report only the in-plane conductivity (σ_||_), which is typically significantly higher than the through-plane conductivity (σ_⊥_), and less relevant for electrochemical applications, in which protons diffuse through the bulk of the material [33,41]. Recently, a similar crosslinking approach was used, in which 20 μm thick CNF membranes were prepared, followed by infiltration with 10 wt.% SSA solution, drying, and hot pressing at 120 °C [27]. This reportedly increased the through-plane proton conductivity from 0.48 mS/cm to 3.17 mS/cm (measured in the wet state, at room temperature), although the membranes were reportedly extremely brittle. Interestingly, both above studies on cellulosic materials reported a decrease in the crystallinity after crosslinking, suggesting that the exposure to SSA partially damages the polymer structure. Recently, a BC-based PEM was crosslinked using tannic acid, resulting in impressive thermal-oxidative stability and proton conductivity of 1.6 mS/cm at 94 °C and 98%RH.

Our work presented an as yet unexplored use of cellulose nanocrystals as a starting material for crosslinking, and took advantage of this stability in the presence of acid. Figure 11 summarizes the membrane formation from water dispersion, crystal packing in the resulting membranes, and the possible proton-conduction mechanisms. Based on the SEM observations, the crystallites in CNC membranes (Figure 11a) become highly aligned after the slow drying of the water dispersion. This behavior is likely driven by the formation of a hydrogen bonding network between the crystallites during the evaporation of water. The aligned proton conductors are thought to be less favorable for utilization in PEMs, because the orientation results in highly anisotropic conductivity, depending on whether it is measured in-plane or through-plane [32] with the latter usually being lower. The activation energy of the proton conduction in the CNC membrane was relatively high, regardless of the humidity, suggesting that the vehicle-conduction mechanism is dominant (Figure 11a), and accounting for the relatively low proton conductivity. Meanwhile, in the SSA-crosslinked CNC membranes, SEM revealed a random orientation of crystallites once the amount of the crosslinker exceeded 5 wt.%, (Figure 11b), and the resulting isotropy of the membrane may contribute to the enhancement of the through-plane conductivity. At a low humidity, the activation energy of proton conduction is similar to that in the CNC membranes but becomes much lower at a higher humidity (>80% RH), suggesting a conduction mechanism similar to Nafion^®^, where hydrated -SO_3_^−^ moieties enhancing both the Grotthuss and hopping pathways for proton conduction (Figure 11b).

This work shows that a rational choice of the nanocrystalline cellulose which is resistant to high concentrations of acid enabled qualitatively better results, compared to other cellulose-based PEMs. The structural level mechanical properties, aqueous stability and the proton conductivity were all improved via crosslinking with SSA. However, the long-term durability of the membranes, and the loss of acidic groups in hot humid conditions could present an obstacle for their application in electrochemical devices. As such, further membrane optimization, testing in a fuel cell environment, and detailed durability tests will be performed in a future study. Although these CNC-based membranes cannot yet compete directly with the benchmark PFSA ionomers, the progress compared to other cellulose-based PEMs is substantial. Research in this field is therefore an important step towards achieving more sustainable PEMs made of renewable materials, especially for smaller scale or short-term applications.

Meanwhile, the existing literature contains many studies where nanocellulose materials are used to fabricate composite PEMs, e.g., by mixing with conventional ionomer materials [12,17]. The inclusion of cellulosic materials in those cases is generally for purposes other than the improvement of the proton conductivity (such as mechanical properties or water uptake). The crucial importance of the results presented in this study is that chemically modified nanocellulose alone can be employed as an ionomer.

## 5. Conclusions

In this work, we demonstrated that CNC membranes crosslinked with SSA via thermal annealing for a short period of time are suitable for use as highly sustainable biopolymer-based PEMs. The CNCs were found to be resistant to acid treatment, allowing the loading of a high proportion of crosslinker without significantly affecting the chemical structure (unlike in the cases of CNFs or micro-fibrillated cellulose). The success of the esterification crosslinking reaction was confirmed by various spectroscopic techniques. Unlike the pristine CNC membranes (which readily redisperse), the crosslinked CNC membranes were mechanically stable in water, even after boiling for three hours. In addition, the crosslinked CNC membranes had better mechanical properties, with >6% elongation at break and a maximum tensile strength up to around 50 MPa. The ion exchange capacity was comparable to Nafion^®^ at high RH (up to 1.5 mmol [H^+^]/g), and the proton conductivity reached 15 mS/cm (a two order of magnitude increase compared to the non-crosslinked samples, and approaching that of Nafion^®^). A remaining issue for future research is a decrease in conductivity after boiling in water, due to de-esterification and subsequent leaching of SSA from the membrane. However, this may be solved by utilization of alternate crosslinking strategies in the future. To the best of our knowledge, this study presents the highest reported through-plane proton conductivity for cellulose-based membranes to date (excluding composites). The strategy of utilizing abundant biomass combined with rational chemical design for the development of high-performing biopolymer PEMs, provides a feasible route for the development of sustainable electrochemical devices, such as fuel cells and electrolyzers.

## Figures and Tables

**Figure 1 membranes-12-00658-f001:**
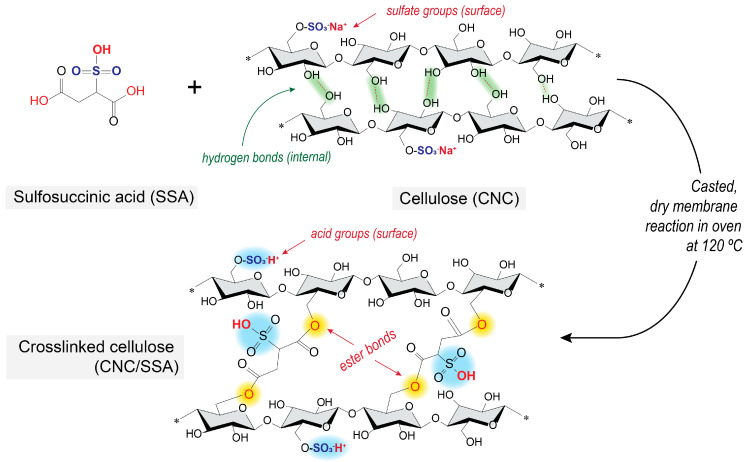
Schematic depicting the crosslinking reaction between cellulose nanocrystals (CNCs) and sulfosuccinic acid (SSA).

**Figure 2 membranes-12-00658-f002:**
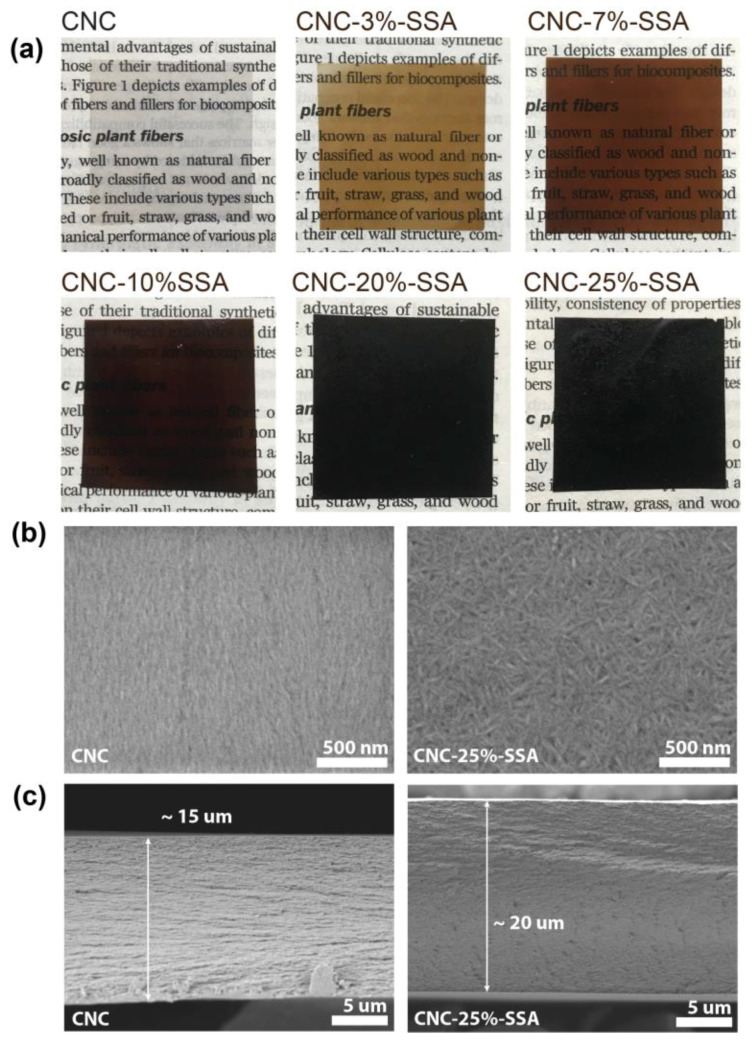
(**a**) Photographs of a pure cellulose nanocrystal membrane (CNC), and five membranes crosslinked using different proportions of sulfosuccinic acid (SSA). The background text is taken from Mohanty et al. [10] to demonstrate the degree of transparency; (**b**) Scanning electron microscopy images of the surface of a CNC membrane (**left**), and a CNC-25%-SSA membrane (**right**); (**c**) Scanning electron microscopy images of the cross-sections of a ~15 μm thick CNC membrane (**left**) and a ~20 μm thick CNC-25%-SSA membrane (**right**).

**Figure 3 membranes-12-00658-f003:**
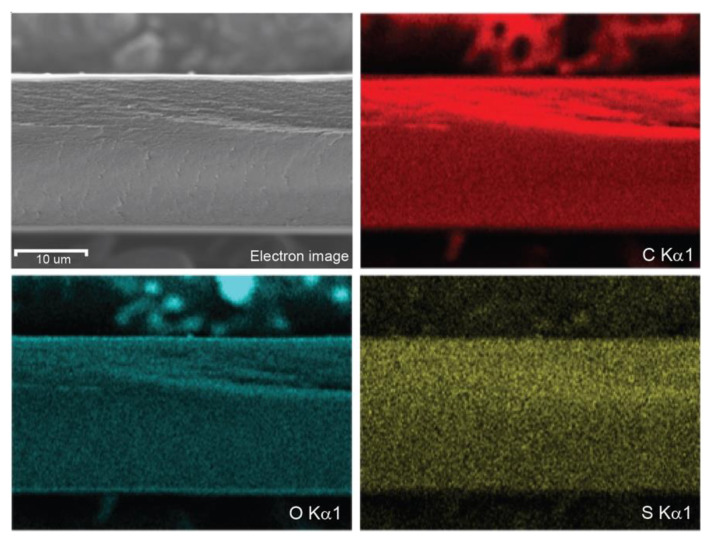
SEM-EDS analysis of a crosslinked CNC-25%-SSA membrane showing weak but homogenous sulfur distribution throughout the membrane, in addition to the main cellulose components (i.e., carbon and oxygen).

**Figure 4 membranes-12-00658-f004:**
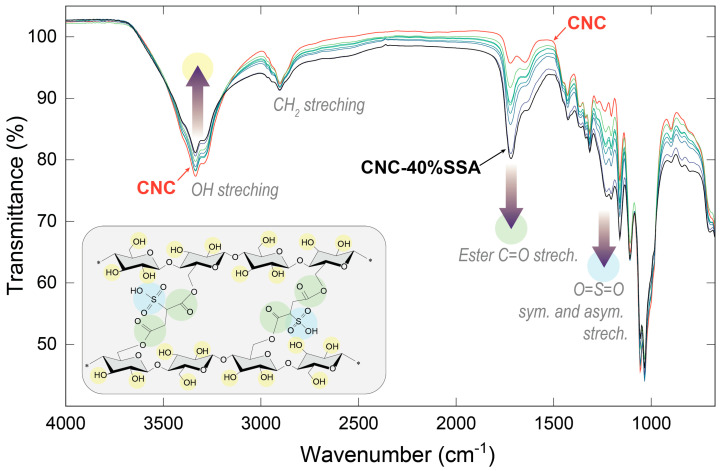
ATR-FTIR spectra of the CNC (red color) and CNC-x%-SSA membranes (green-blue colors) crosslinked with the different amounts of sulfosuccinic acid (where x = 0 to 40%).

**Figure 5 membranes-12-00658-f005:**
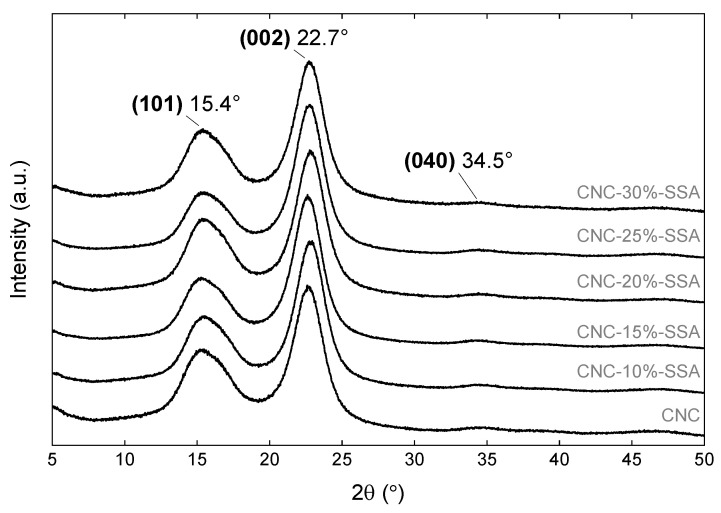
X-ray diffraction spectra of the CNC and CNC-x%-SSA membranes (where x = 0 to 30%).

**Figure 6 membranes-12-00658-f006:**
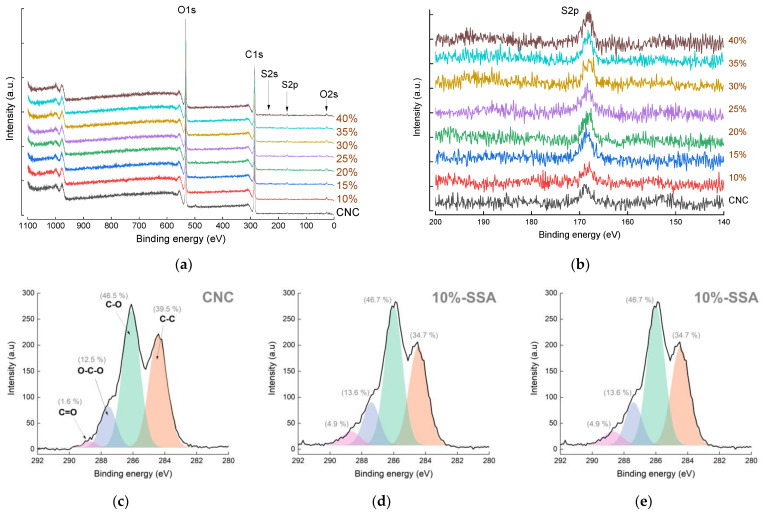
XPS spectra of cellulose nanocrystal (CNC) and crosslinked CNC-x%-SSA membranes (where x = 0 to 30%): (**a**) wide scan spectra; (**b**) S 2p region; and (**c–e**) C 1s region of the CNC, CNC-10%-SSA, and CNC-25%-SSA membranes, respectively.

**Figure 7 membranes-12-00658-f007:**
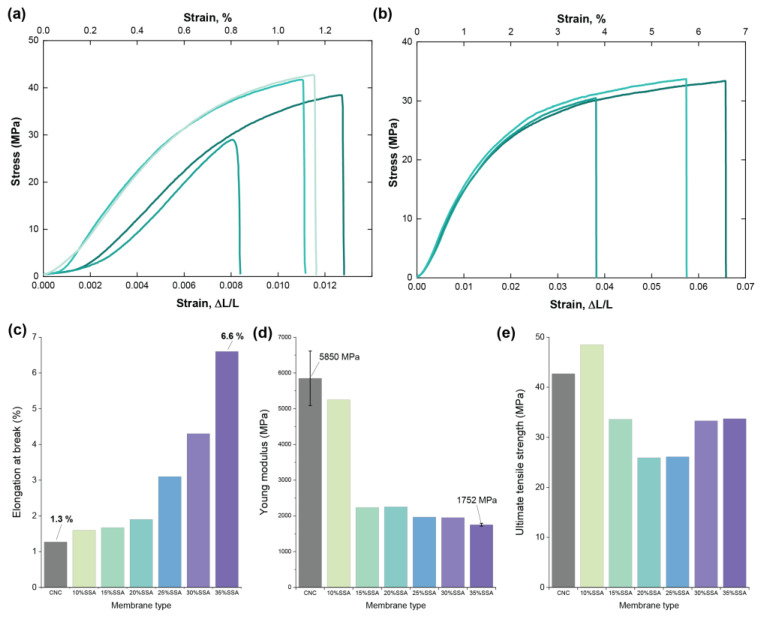
Mechanical properties of the fabricated membranes estimated in a tensile test: stress–strain curves for (**a**) CNC and (**b**) CNC-35%-SSA membranes; (**c**) elongation at break; (**d**) Young’s modulus; and (**e**) ultimate tensile strength of membranes prepared with up to 35% of SSA in the membrane forming solution.

**Figure 8 membranes-12-00658-f008:**
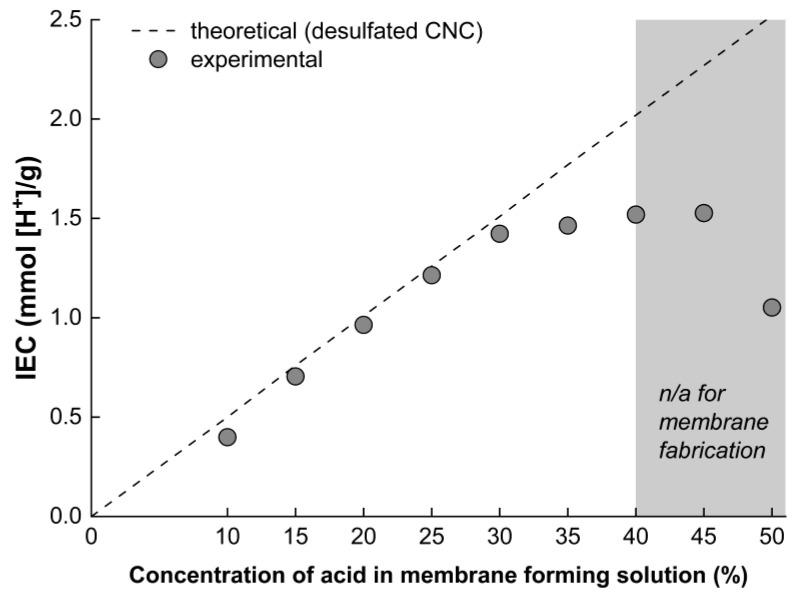
Variation of the IEC of the CNC-x%-SSA membranes with the proportion of SSA.

**Figure 9 membranes-12-00658-f009:**
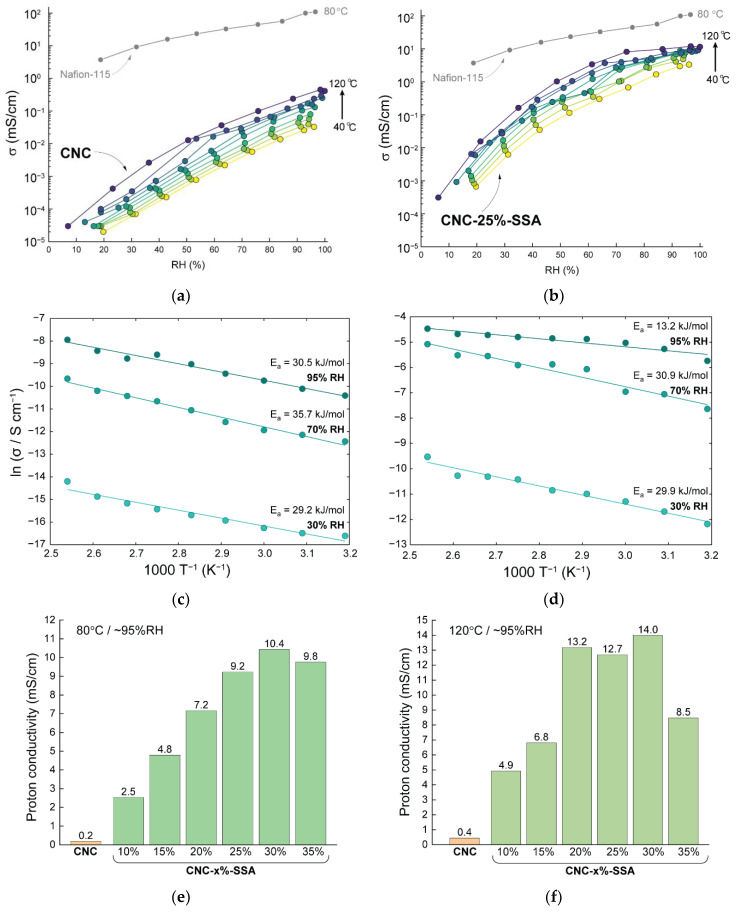
Dependence of the proton conductivity of (**a**) CNC and (**b**) CNC-25%-SSA membranes on the relative humidity at different temperatures (40 to 120 °C) in comparison to commercially available Nafion-115 membrane. Arrhenius plots for (**c**) CNC and (**d**) CNC-25%-SSA membranes measured at 30, 70 and 95% RH. Proton conductivity of CNC and CNC-x%-SSA membranes as a function of the proportion of SSA at (**e**) 80 °C and (**f**) 120 °C, at maximum relative humidity (~95% RH).

**Figure 10 membranes-12-00658-f010:**
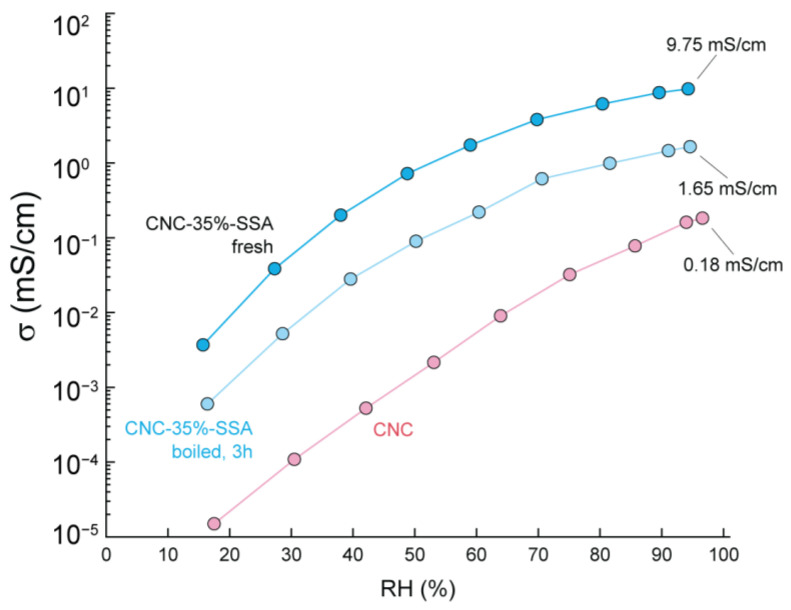
Through-plane proton conductivity of CNC-35%-SSA membrane as prepared (blue) after 3 h of boiling in deionized water (light blue) compared to pristine CNC as a function of the relative humidity at measured at 80 °C (red).

**Figure 11 membranes-12-00658-f011:**
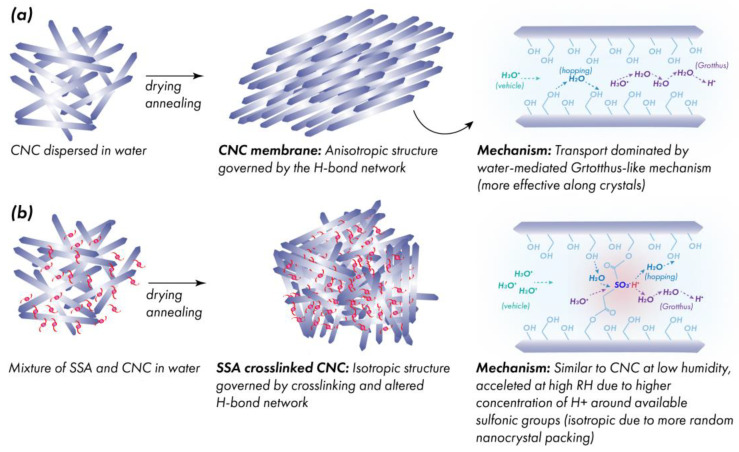
Mechanisms of membrane formation and proton conduction in (**a**) pristine and (**b**) crosslinked CNC.

**Table 1 membranes-12-00658-t001:** Mechanical properties of the cellulose-based membranes prepared via crosslinking with sulfosuccinic acid compared to the literature reported studies.

Membrane Description in Original Work	Type of Cellulose, Fabrication Procedure	Young’s Modulus (MPa)	Tensile Strength at Break ^1^ (MPa)	Elongation at Break (%)	Reference
Cellulose/SSA 5%	Microcrystalline powder (Sigma Aldrich), dissolved in DMSO (stirring), mixed with SSA (stirring 4 h), cast on Teflon-coated glass, dried 80 °C/48 h, annealed 120 °C/3 h Washing: not reported	2.075	2.568	0.808	[24]
Cellulose/SSA 10%		4.457	4.772	0.955	
Cellulose/SSA 15%		5.649	5.915	0.955	
Cellulose/SSA 20%		5.55	2.471	2.246	
Cellulose/SSA 25%		0.606	0.298	2.034	
Cellulose/SSA 30%		1.057	0.498	2.121	
NC	Cellulose nanofibers (prepared from Para-rubber wood sawdust). Membranes fabricated by 0.82% H_2_O suspension pressure filtration, following by hot pressing at 120 °C/1 h. Membranes were soaked in aq. SSA solutions (0.1–10wt.%)/3 days, followed by hot pressing 120 °C/1 h	1.36	0.29	4.66	[27]
NC-0.1SSA		0.31	0.19	1.59	
NC-1SSA		0.15	0.36	0.46	
NC-3SSA		0.09	0.86	0.11	
NC-5SSA		0.08	0.91	0.09	
CNC	Cellulose nanocrystals (CNC, Celluforce Inc. Montreal, QC, Canada), dissolved in H_2_O (mechanical blender), mixed with SSA (ultrasonication 5 min), cast on PTFE Petri-dish, dried 35 °C/48 h, annealed 120 °C/10 min Washing: thorough in DI water, three cycles	5850	42.7	1.27	This work
CNC-10%-SSA		5249	48.5	1.6	
CNC-15%-SSA		2233	33.6	1.67	
CNC-20%-SSA		2252	25.9	1.9	
CNC-25%-SSA		1965	26.1	3.1	
CNC-30%-SSA		1950	33.3	4.3	
CNC-35%-SSA		1752	33.7	6.6	
Nafion-212	Commercially available	245	14	50	This work

^1^ For all samples in this work maximum tensile strength was achieved at break, i.e., ultimate tensile strength is equal to tensile strength at break.

**Table 2 membranes-12-00658-t002:** Literature reported IEC and proton conductivity in cellulose-based membranes that satisfy the sustainability requirements, i.e., fabricated via chemical means (e.g., surface modification, chemical crosslinking) using sustainable materials as a source.

Membrane Material	Ion Exchange Capacity, mmol [H^+^]/g	Measurement Approach	Proton Conductivity, mS/cm	Measurement Condition	Reference
CNC	*n/a*	Through-plane	4.6	120 °C, 100% RH	[16]
CNF	*n/a*		0.05	100 °C, 100% RH	
S-CNF			2	120 °C, 100% RH	[18]
Cellulose/SA 5%	0.12	In-plane	0.9	20 °C (in DI water)	[26]
Cellulose/SA 10%	0.09		2.4		
Cellulose/SA 15%	0.27		8.0		
Cellulose/SA 20%	0.52				
Cellulose/SA 25%	0.42		15.0		
Cellulose/SA 30%	0.53		23		
			40	85 °C (in DI water)	
CNF	0.005	Through-plane	0.48	r.t. (in DI water)	[27]
CNF—0.1SSA	0.006		0.37		
CNF—1SSA	0.010		0.16		
CNF—3SSA	0.033		0.12		
CNF—5SSA	0.043		0.73		
CNF—10SSA	0.069		3.17		
H-CNF-600 (14 um)	*n/a*	In fuel cell	1.4	30 °C, 95% RH	[38]
H-CNF-1550 (14 um)	*n/a*		1.5		
H-CNF-600 (24 um)	*n/a*		1.2		
H-CNF-1550 (24 um)	*n/a*		1.4		
BC	*n/a*	Through-plane	0.063	94 °C, 98% RH	[39]
BC/Fuc_50			0.76	80 °C, 98% RH	
BC/Fuc_75			0.78	80 °C, 98% RH	
BC/Fuc_75			1.6	94 °C, 98% RH	
CNC	*n/a*	Through-plane	0.4	120 °C, ~96% RH	This work
CNC-10%-SSA	0.399		4.8	120 °C, ~96% RH	
CNC-15%-SSA	0.705		7.5	120 °C, ~96% RH	
CNC-20%-SSA	0.964		11.6	120 °C, ~96% RH	
CNC-25%-SSA	1.214		12.7	120 °C, ~96% RH	
CNC-30%-SSA	1.423		14.0	120 °C, ~96% RH	
CNC-35%-SSA	1.464		10.1 *	100 °C, ~96% RH	

SSA: Sulfosuccinic acid, H-CNF-*n*: carboxylated CNF prepared by TEMPO-mediated oxidation of cellulose pulp, number *n* indicates the surface charge density (mmol/g), BC: bacterial cellulose produced by gluconacetobacter sacchari, Fuc_x: fucoidan from fucus vesiculosus seaweed, x indicates the amount of fucoidan in the membrane (wt.%), * The maximum of conductivity achieved at 100 °C.

## Data Availability

Data generated in this study is contained within the article and Appendix A.

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
