# Peer review of "Cellulose Nanocrystals Crosslinked with Sulfosuccinic Acid as Sustainable Proton Exchange Membranes for Electrochemical Energy Applications"

_membranes, 2022, doi:10.3390/membranes12070658_

Round 1

Reviewer 1 Report

In this paper, nanocrystalline cellulose is used to prepare proton exchange membranes (PEMs) and their properties were evaluated by various ex-situ characterization techniques, including FTIR spectroscopy, EDX spectroscopy, XRD, SEM, IEC and proton conductivity. Although the authors claimed that the cross-linked cellulose membranes results in enhanced proton conductivity, the properties of the synthesized membranes still are not superior/comparable to commercial Nafion and their chemical stability is a big concern. Considering the structure of the synthesized membrane is relative new, we still would like to suggest this manuscript be accepted for publish in Membranes after the following issues have been addressed.

11.    The introduction section is too long and need to be condensed and reorganized.

22. Page 4, line 165, “the chemical structure likely contains some -SO3+ functionalities”, it can be approved by FTIR, XPS.

33.  Page 6, line 246,   “the cross-linked membranes display a systematic change in color to dark brown and then black”.  Why were the color of the cross-linked membranes changed?

44.    Page 9, line 316, “The results are remarkably similar for all samples, indicating that the crosslinking reaction does not affect the crystalline structure of CNC….” Please compare the peak area between (101) and (002) to see whether the peak ratio is changed. 

55.   Figure 6b, the difference in S2p peaks of XPS between CNC and CNC-x%-SSA is not obvious. XPS is not an effective approach to identify the difference in chemical structure between samples. NMR? Raman spectroscopy?  Find other approaches to characterize your cross-linked samples?  

66.   Figure 11, the duration for stability test is too short. Could you please extend the testing time from 3 h to a week? The temperature can be change to 80 or 90°C according to the fuel cell testing conditions. Please also check conductivity, mechanical properties and IEC as a post analysis.

Author Response

The PDF file with replies is attached.

Reviewer 2 Report

The paper focused on the development of Nanocellulose-based proton exchange membrane with sulfonic acid crosslinker. The proposed approach allows improving the mechanical robustness, water-stability, and proton conductivity of the obtained membranes, which makes them promising to apply in electrochemical devices such as PEM fuel cells, etc.

This work is meaningful, which can be published in Membranes journal after a minor revision by addressing the following comments:

1. Authors proposed the obtained membranes for applications such as fuel cells and water electrolyzers. The proposed data proved Nanocellulose-based proton exchange membrane to be a promising alternative to Nafion (in terms of mechanical properties, proton conductivity and thermal stability even at 120 °C). But what about the specific stability issues related to aforementioned applications? Fuel cell cathode and water electrolyzer anode operates at relatively high anodic overvoltage, which is considered as a significant factor hindering the durability of Fuel cells and water electrolyzers. So, authors should discuss if the proposed membranes could withstand anodic polarization, or their potential.

3. The Figure S4 in Supporting material have no reference in the main text. So, authors should at least add the reference or shortly discuss the data given in figure.

3. The manuscript must be carefully checked to avoid obvious mistakes (page 3 line 131 …SSA as also….; page 9 line 332 …moiety35,…).

4. The literature survey might be enlarged using some relevant references related to the cellulose membranes applied to practical use in electrochemical devices, such as 10.1039/C9SE00381A, 10.1016/j.carbpol.2017.12.074, 10.1016/j.ijhydene.2019.09.194, 10.1016/j.electacta.2017.02.145

Author Response

The PDF file with replies is attached.

Reviewer 3 Report

In this manuscript, nanocrystalline cellulose is used to prepare PEMs. The sulfonic acid crosslinker is used to simultaneously improve mechanical robustness, water stability, and proton conductivity. The enhanced proton conductivity up to 15 mS/cm can be reached. It can be concluded that nanocellulose can act as a sustainable and low-cost alternative to the current commercial perfluorosulfonic acid ionomers for applications in fuel cells and electrolyzers.

I may give a minor revision due to further improvements that are needed by addressing my comments made below.

(1) For the Keywords, I suggest adding fuel cell, ‘sulfonic acid groups, and ‘low-cost alternative’ to attract a broader readership.

(2) There are many grammar problems. It is suggested that the full text be greatly improved.

(3) Line 53, Currently, Nafion® is the benchmark polymer used in PEMs. This is a sulfonated fluoropolymer with high proton conductivity (~100 mS/cm) which has dominated the market since its discovery in the late 1960s [5]. However, Nafion® is extremely expensive...

Firstly, why does Nafion have so high proton conductivity? It should be explained briefly, especially about the significant phase separation between the hydrophobic and hydrophilic domains when hydrated provides a relatively larger channel for the proton transportation [Solid State Ionics 319 (2018): 110-116]. This is also helpful to explain why in this work sulfonic acid crosslinker is used.

Secondly, except being hard to recycle, the low permselectivity is also a drawback for Nafion membranes, no matter for the DMFC (high permeation of methanol) or flow batteries (high permeation of active species), which indicates Nafion is not a good barrier layer to prevent active species crossover at all [Materials Today: Proceedings 35 (2021): 344-351; Electrochimica Acta 378 (2021): 138133]. This issue should also be mentioned and one concern should also be paid to the permselectivity of the obtained materials in this work, not only with good proton conductivity.

(4) Line 59, Regarding sustainability and cost, a promising class of new materials are sulfonated hydro-carbon PEMs... still rely on the extraction of fossil fuels, and expensive fabrication processes. This is one side, but also another side should be addressed, that is the long-term chemical stability and polymer degradation issue due to the backbone is not strong enough compared to Nafion.

(5) Page 3, this page lists different membrane components and corresponding performance data in many literatures, but they are too scattered and are all described in words. Is it possible to summarize them in a table? In this way, readers can compare more intuitively.

(6) Why the water uptake and swelling ratio is not measured for the membrane? For the physical-chemical parameters, they are also very important besides IEC. And also, why the chemical stability by Fenton reagents are not carried out? Proton conductivity is one side, but chemical stability and permselectivity are also very important.

(7) In Figure 4, it is nearly impossible to distinguish the sample name and colour. This information should be supplied. And ~1200, it should be ‘symmetrical’ rather than asym stretch', see Table 5 of [Electrochimica Acta 309 (2019): 311-325; 10.1016/j.memsci.2013.09.058].

(8) Why there is no in-situ polarization test for the fuel cells? This is very important and it should be compared with Nafion membranes, to see the difference during fuel cell applications. 

Author Response

The PDF file with replies is attached.

Round 2

Reviewer 1 Report

 Accept in present form.